# Ask, Acquire, Understand: A Multimodal Agent-based Framework for Social Abuse Detection in Memes

## Abstract

Memes serve as a powerful medium of expression in the digital age, shaping cultural discourse and conveying ideas succinctly and engagingly. However, their potential for social abuse highlights the importance of developing effective methods to detect harmful content within memes. Recent studies on memes have focused on transforming images into textual captions using large language models (LLMs). However, these approaches often result in non-informative captions. Furthermore, previous methods have only been tested on limited datasets, providing insufficient evidence of their robustness. To address these limitations, we present a multimodal, agent-based framework designed to generate informative visual descriptions of memes by asking insightful questions to improve visual descriptions in zero-shot visual question-answering settings. Specifically, we leverage an LLM as agents with distinct roles and a large multimodal model (LMM) as a vision expert. These agents first analyze the images and then ask informative questions related to potential social abuse in memes to obtain high-quality answers about the images. Through continuous discussion guided by instructional prompts, the agents gather high-quality information while repeatedly acquiring image data from the LMM, which helps detect social abuse in memes. Finally, the discussion history and basic information are classified using the LLM to obtain the final prediction results in a zero-shot setting. Experimental results on a meme benchmark dataset sourced from 5 diverse meme datasets, comprising 6,626 memes spanning 5 tasks of varying complexity related to social abuse, demonstrate that our framework outperforms state-of-the-art methods, with detailed comparative analysis and ablation studies, further validating its generalizability and ability to retrieve more relevant information for detecting social abuse in memes.

Disclaimer: This paper contains content that may be disturbing to some readers.

## CCS Concepts

• **Information Extraction → Multimodality**.

## Keywords

Meme Analysis, Social Abuse, Multimodal

**ACM Reference Format:**
Anonymous Author(s). 2018. Ask, Acquire, Understand: A Multimodal Agent-based Framework for Social Abuse Detection in Memes. In *Proceedings of Make sure to enter the correct conference title from your rights confirmation emai (Conference acronym 'XX).* ACM, New York, NY, USA, 11 pages. https://doi.org/XXXXXXX.XXXXXXX

## 1 Introduction

The rapid development of social media has greatly transformed the way information is generated, shared, and consumed, surpassing any previous medium. Unfortunately, this growth has also led to a significant rise in the misuse of memes online [2]. Memes, which typically combine images and texts to convey humor or satire, have become a popular mean of spreading information and ideas across social media platforms. While many memes are innocuous and serve as a form of entertainment, others can be detrimental, propagating misinformation, encouraging hatred, or causing offense [41].

Detecting social abuse in memes presents a significant challenge due to the subtlety of their content, where the underlying meanings are not immediately apparent in the text and images, making it notably difficult to evaluate their negative impact [37]. Past studies on detecting memes have primarily relied on conventional pretrained encoders for extracting image and text features. These studies have also focused on developing novel techniques for fusing multimodal data representations [41] and reducing the disparity between modalities to capture semantic and contextual information. Although fusion-based models have demonstrated improved performance, they may not be the most suitable option for analyzing memes containing hateful content, as the role of text in memes differs from that of image captions, and the text and image may convey different meanings [24]. Despite their advances, these methods are often challenged by the gap between heterogeneous modalities, resulting in a semantic gap and insufficient learning in fusion-based approaches [48].

Recent studies [8, 18, 19] on detecting memes have transformed the task of multimodal meme detection by framing it as an unimodal masked language modeling problem. These recent methods first generate captions for meme images using an image caption model. However, despite achieving state-of-the-art (SOTA) performance, these prompt-based methods are heavily affected by the quality of image captions [8]. When image captions merely provide generic descriptions, crucial details of individuals may be omitted, which are vital for detecting social abuse in memes.

In parallel, recent research [7, 23, 36, 44, 46] has highlighted the impressive zero-shot learning abilities of large language models (LLMs) that are fine-tuned to follow instructions. These LLMs can perform new tasks in a zero-shot manner when presented with well-crafted instruction prompts. Despite the significant progress, their effectiveness in providing useful information heavily relies on the quality of the prompts received. Essentially, these models depend on humans to ask insightful questions that can direct their generation of informative answers. We argue that if we have an automatic questioning machine that asks informative questions, the human questioners can be replaced, and the computational models can be guided to provide valuable knowledge automatically (see Figure 1).

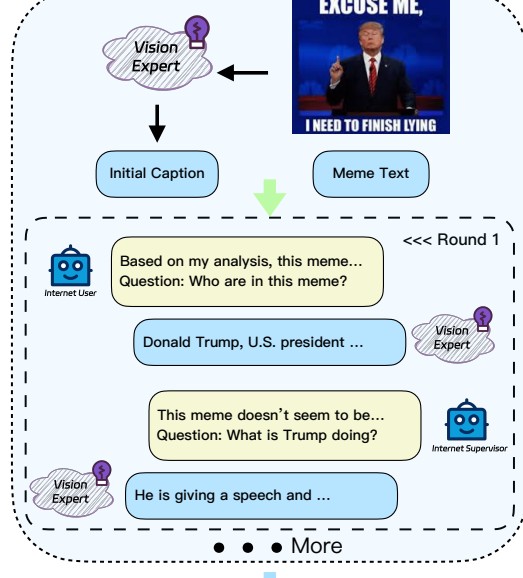

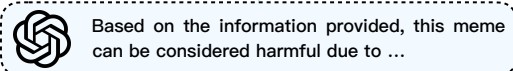

**Figure 1: Example of multi-agent chat. Starting with the initial caption and meme text provided, agents can analyze the current information and decide what to ask about the image to get a better understanding. This process will repeat multiple times.** *The dialogue is incomplete due to space limit.*

Another limitation of current methods is their lack of generalizability. While these methods may achieve impressive performance, they often exhibit poor robustness to real-world scenarios where social abuse varies. This lack of generalizability hampers their practical application in detecting and mitigating social abuse across diverse social media platforms and communities.

To address the limitations of previous work, we propose a novel multi-agent framework that leverages the power of LLMs and LMMs to generate descriptions of memes through insightful agent discussions[1]. Our approach involves creating two distinct agents with unique roles using GPT-3.5 to pose targeted questions about the visual content of memes and receiving answers from a vision expert powered by LLaVA-1.6. Through continuous discussion guided by instructional prompts, these agents obtain high-quality information that helps capture social abuse in memes. Finally, we perform zero-shot classification using an LLM to generate the final prediction results by utilizing the question-answer history and basic information extracted from the meme, such as text and basic image captions. Our main contributions are as follows:

- We introduce a novel multimodal multi-agent framework to generate informative meme descriptions by asking insightful questions and enhancing visual descriptions in zero-shot settings. To the best of our knowledge, we are the first to apply a multi-agent approach to detecting social abuse in memes.

---

[1]Our approach offers flexibility by allowing the substitution of any LLM and LMM.

- We leverage an LLM as two agents and an LMM as a vision expert to ask targeted questions and obtain high-quality answers. Specifically, the agents continuously discuss through instructional prompts, gathering informative captions from the LMM. Finally, the LLM leverages the generated discussion history from the previous step to classify and produce the final predictions.
- Experimental results on the memes benchmark dataset, collected from five meme datasets comprising 6,626 memes across five tasks of varying complexity related to social abuse, show that our framework outperforms SOTA methods and is generalizable in identifying social abuse in memes.

## 2 Related Works

### 2.1 Meme Detection

Early studies extract features from image and text and combine them to understand the meme's overall meaning. Fusion methods, including early fusion [20, 34] and late fusion [21], combine representations from various modalities, allowing the model to learn a unified representation. Additionally, MOMENTA [39], a multi-modal framework, uses a cross-modal attention mechanism to address alignment challenges. However, differences between modalities often lead to missing information in the learned representation.

In recent studies, researchers convert all modalities into text to bridge the gap between different modalities. In single-modality classification tasks, one critical objective is to extract sufficiently rich information from the images. PromptHate [8] leverages a VLM to generate an image caption. Besides, PromptHarm [19] introduces attributes as higher-level concepts for images. Pro-Cap [7] manually designed several questions to get related information from memes by BLIP-2. After that, these textual features were then fed into a Pre-trained Language Model (PLM) to produce the final classification results. Recently, Ji et al. [18] presented CapAlign, where they prompted a ChatGPT to ask questions to BLIP-2 and used the dialogues to generate image captions for harmful meme analysis.

Compared to previous works like CapAlign, which uses an uncontrolled automatic question-asking process, our approach employs a multi-agent framework that continuously summarizes information during dialogue, improving the quality of visual descriptions in zero-shot settings. (i) Our method enhances caption generation using a more interactive and dynamic process, unlike CapAlign, which lacks iterative refinement; (ii) while CapAlign is tested on limited datasets, our method offers a robust solution across five tasks related to identifying memes with social abuse content; (iii) CapAlign operates in a supervised manner, requiring training datasets, whereas our approach is fully zero-shot, detecting social abuse memes without any prior training.

### 2.2 Zero-shot Applications of LLMs and LMMs

Recent research findings [11, 36, 45] have showcased the exceptional capabilities of LLMs like GPT-3 [5] and GPT-4 [1] in handling a wide range of tasks defined by prompts without training. Additionally, the Chain of Thought (CoT) method [46], which progressively guides the problem-solving process within the prompts, significantly enhances their problem-solving performance.

The progress in LLMs has been a key catalyst for advancing LMMs, leading to pretrained LMMs showing promising capabilities

**Figure 2: Architecture of our framework. The meme text and initial caption are used to initiate the multi-agent chat. Through agent discussion (*Ask*), informative information is acquired from the vision expert (*Acquire*). The QA histories and basic meme information assist the summary assistant in understanding the meme (*Understand*) before generating the final result.**

in zero-shot question answering. In visual question answering, the Q-Former within BLIP-2 [26] translates visual features into tokens that can be directly understood by a frozen LLM, playing a crucial role in understanding images. Moreover, CogVLM [43], Qwen-VL [4] and the LLaVA series [29–31], showcase encouraging results in coordinating visual encoders with LLMs through instruction tuning performed on high-quality instruction-tuned datasets synthesized by ChatGPT. Compared to previous works, we use the GPT-3.5 and LLaVA-1.6-13B model to establish a multimodal agent-based framework, leveraging their zero-shot capabilities.

### 2.3 Multi-Agent

Different agents can autonomously communicate and negotiate, enabling them to collectively address complex tasks by acting as communicators [14, 16, 33]. These agents, often embodied by LLMs, are designed for effective interactions with other agents or human users through natural language. Various studies have explored the aspects of communicative agents. Studies [13, 27, 47] employed static debate-style participation among LLMs to enhance reasoning. [35] enabled agents to engage in multi-round interactions within a dynamic framework.[25] proposed a cooperative agent framework, termed role-playing, allowing agents to collaborate on complex tasks autonomously. [32] developed a sandbox environment consisting of 25 separate virtual entities, each with assigned role descriptions and memory systems. [27] and [13] also utilized multi-agent debate frameworks in other scenarios, such as translation and arithmetic problems, achieving improved outcomes. To the best of our knowledge, no prior work has used multi-agents for meme analysis, in particular, to identify social abuse in memes. We fill this gap and introduce a novel framework that uses multi-agent to simulate different roles involved in the meme propagation process and obtain more effective and comprehensive information for meme analysis.

### 3 Method

**Overview:** Our proposed method (Figure 2) consists of three modules: (i) a vision expert module for acquiring image information, (ii) a multi-agent module that contains two agents with different role settings, and (iii) a summary module that analyses the chat history for final classification.

### 3.1 Vision Expert Module

We found that current open-source LMMs [4, 29, 31, 43] primarily excel at image-related instruction following due to their minimal exposure to text-only datasets, resulting in poorer performance in text-only interactions. To address this, we chose to employ an LLM with robust logical reasoning capabilities as the primary component for our agents while leveraging an LMM as a vision expert tool to gather image information.

Specifically, we employed LLaVA-1.6-13B as the vision expert model for tool invocation and defined a specific format for tool calls. The agent's question about the meme is placed between *<question>* and *</question>* tags to locate and extract the question accurately. The question requests to LMM are constructed as follows:

```
Only answer what is asked: {Question}
```

Our tests revealed that adding the prefix prompt effectively mitigates the LMM's hallucinations and ensures that its output answers adhere closely to the question. Finally, the vision expert tool's output is fed back to the agents' chat discussion session as additional meme information.

### 3.2 Multi-Agent

**Question and Answer:** Inspired by previous studies [9, 18, 25], we introduce a multi-agent system that leverages the role-playing capability of LLMs to automatically ask questions about a meme. Specifically, we employ LLM agents to discuss the information of a meme, ask questions about it, and ultimately acquire more accurate, informative, and relevant information about the meme.

**Role Definition:** We utilize GPT-3.5 to play two roles: *Internet User* and *Internet Supervisor*. The Internet User views a meme's information from the perspective of an ordinary Internet user, while the Internet Supervisor examines it from the standpoint of an Internet supervisor. These two agents engage in discussion, ask questions, and continuously update the meme's image information through the vision expert to initiate the next round of discussion. In the *Role definition* (see Table 6 in the Appendix A.1) of these two agents, [role] represents Internet User or Internet Supervisor, and [adj] represents the specific category of identifying social abuse tasks, such as *sarcastic* or *harmful*.

**Discussion Process:** To ensure that the two agents remain focused on the meme during the discussion, we designed a series of instructions that guide the discussion to follow the intended flow. After initiating the discussion, agents take turns to generate their responses as instructed. As a result, we have the following expectations from the discussion process of our farmework:

- Their task is to acquire as much image information about the meme as possible for the final social abuse detection.
- Prevents their discussion from deviating from the meme image to avoid introducing excessive hallucinatory information.
- Specify the method for agents to obtain image information from the tool and impose limitations on their questioning of it.

To provide a topic for the agents' first discussion, we initialized some basic information, including the text on the meme and the initial caption of the meme image obtained through the vision expert. This information is added to the agent's definition. Further, we use agents' discussion history to connect the meme's basic information and the agent's definition. When it is the first round of dialogue, i.e., there is no discussion history, we simply use *"This is the first round."* to replace it. Finally, we use *"Now it is your time to talk"* to prompt the agents to begin their discussion session. Refer to Appendix A.1 for details related to the prompts designed for discussion process.

## 3.3 Summary and Classification

After several rounds of agent dialogue, we obtain their discussion history and supplementary image information that is more relevant to the social abuse of the meme based on the discussion. At this point, we similarly use GPT-3.5 to define a summary module to organize all the information and derive the final classification results. We only utilize questions and answers about the meme image within the discussion history to avoid biases in agents' speech. In order to enable the LLM to have a clearer understanding of each task, we add corresponding definitions for each task to obtain more accurate analysis results (see **"Definitions"** in Section 4 for details). Then, we employ the CoT approach to analyze these historical records before deriving the classification results. The first step is to have the model analyze the factors related to social abuse in the meme based on the task definition. The second step is to output the classification using the specified format based on the analysis results from the first step. Please refer to Appendix A.1 for details related to the prompts designed for the chat history analysis and classification module.

## 4 Experimental Settings

**Datasets:** We utilized the GOAT-bench dataset [28], which consists of 6,626 memes across 5 distinct tasks related to social abuse: hateful, misogynistic, offensive, sarcastic, and harmful content. The GOAT dataset is curated from 5 diverse memes datasets, including the FHM dataset [22] for evaluating hatefulness, the MAMI dataset [15] for misogynistic content, the MultiOFF dataset [42] for offensive material, the MSD dataset[6] for sarcasm, and the Harm-C and Harm-P datasets [38, 40] to examine harmfulness. Table 1 provides detailed descriptions of each task along with the statistics of the datasets used in our experiments.

**Metrics:** The evaluation metrics employed in our study were commonly utilized in previous research with similar objectives [28].

**Table 1: Statistics of the GOAT-bench dataset**

| Dataset sourced from | Tasks | Label Distribution | | Total |
|---|---|---|---|---|
| FHM [22] | Hatefulness | Hateful | 750 | 2000 |
| | | Non-hateful | 1250 | |
| MAMI [15] | Misogyny | Misogynistic | 500 | 1000 |
| | | Non-misogynistic | 500 | |
| MultiOFF [42] | Offensiveness | Offensive | 305 | 743 |
| | | Non-offensive | 438 | |
| MSD [6] | Sarcasm | Sarcastic | 910 | 1820 |
| | | Non-sarcastic | 910 | |
| Harm-C and Harm-P [38, 40] | Harmfulness | Harmful | 444 | 1063 |
| | | Non-harmful | 619 | |

Specifically, we used Accuracy and Macro-averaged F1 Score to evaluate the effectiveness of our method.

**Experimental Setup:** We tested our baselines and proposed method using PyTorch on a server with 8 NVIDIA V100 GPUs, each with 32GB of VRAM and CUDA version 12.1.0 installed. For the LLaVA model in our method, we used the 13b version of the Huggingface implementation with a Vicuna-1.5-13b backbone and a vision encoder of 303.5M parameters. We used GPT-3.5-Turbo API as the LLM backbone for the agents and the Summary module. For the baselines, we used Huggingface checkpoints for convenience of implementation and reproduced all results ourselves for consistency.

**Parameters:** We used LLaVA-V1.6-13b as our vision expert. For the LLaVA model, we specifically set *do_sample* to false to get the same response. In the multi-agent chat module, we used GPT-3.5-Turbo to power both agents involved, and we followed the default setting of the OpenAI API to enable creative thinking and get high-quality questions about the meme. Specifically, all *GPT-3.5* used are *gpt-3.5-turbo-0125* version. For agents, we set 2 *rounds of discussion*[2], and all prompts are hand-crafted, which can be found in the Method section. In the Summary module, the Temperature of GPT is set to 0 for classification accuracy. The baseline models we used all have *do_sample* set to false for reproducibility, and their prompts stay the same as proposed in [28] with no modification except additional *Only answer yes or no.* for classification.

**Definitions:** We used the following definitions[3] in our prompts mentioned in Table 5 of Appendix A.1:

- *Hateful* content can be defined as speech or material that has the potential to cause emotional discomfort to individuals targeting ethnicity, gender, or disability and so on with dehumanizing speech and mockery of hate crimes.
- *Misogynistic* content can be defined as speech or material that has the potential to cause emotional discomfort to women with sexism or hate, involving shaming, stereotyping, objectification, violence and so on.
- *Offensive* content can be defined as speech or material that has the potential to cause emotional discomfort to any individual or politician with personal attack, homophobic abuse, racial abuse, or attack on minorities and so on.
- *Sarcastic* content can be defined as speech or material that uses irony or humor through incongruous text and imagery.

---

[2]Empirical test showed that using two rounds of discussion achieved better performance

[3]Definitions can be changed for better alignment.

**Table 2: Results: Proposed v/s the baselines. * indicates that our method achieved a significant ($p < 0.05$) performance improvement over second best approach (underlined) under Mann–Whitney U test.**

| Model | Hatefulness | | Misogyny | | Offensiveness | | Sarcasm | | Harmfulness | | Overall | |
|---|---|---|---|---|---|---|---|---|---|---|---|---|
| | Acc. | F1 | Acc. | F1 | Acc. | F1 | Acc. | F1 | Acc. | F1 | Acc. | F1 |
| CogVLM-17B | 62.55 | 59.80 | 57.20 | 48.75 | 44.41 | 38.29 | 50.05 | 33.46 | 58.61 | 58.54 | 54.56 | 47.77 |
| LLaVA-1.5-13B | 62.80 | 61.25 | 64.30 | 59.97 | 48.05 | 43.86 | 52.03 | 37.70 | 62.44 | 64.21 | 58.32 | 53.40 |
| LLaVA-1.6-13B | 57.10 | 56.98 | 51.80 | 37.68 | 46.03 | 40.09 | 50.00 | 33.33 | 52.30 | 49.65 | 51.45 | 43.54 |
| InstructBLIP-13B | 59.60 | 57.74 | 70.40 | 70.10 | 54.71 | 54.70 | 55.93 | 47.09 | 59.09 | 58.38 | 59.95 | 57.60 |
| miniGPT4-13B | 61.85 | 42.16 | 52.10 | 38.74 | 59.89 | 56.15 | 55.22 | 54.60 | 58.04 | 54.33 | 57.42 | 49.20 |
| Qwen-VL-10B | 65.30 | 58.33 | 72.20 | 71.90 | 60.30 | 60.30 | 52.30 | 40.10 | 60.96 | 50.75 | 62.21 | 56.28 |
| OpenFlamingo-9B | 38.25 | 29.32 | 53.70 | 51.23 | 41.05 | 29.84 | 52.03 | 40.15 | 43.37 | 33.53 | 45.68 | 36.81 |
| MMGPT-9B | 62.50 | 38.46 | 50.00 | 33.33 | 56.80 | 45.69 | 53.96 | 50.73 | 55.69 | 44.81 | 55.79 | 42.60 |
| Fuyu-8B | 37.85 | 28.74 | 49.90 | 33.64 | 46.09 | 43.22 | 48.96 | 36.77 | 48.96 | 47.78 | 46.35 | 38.03 |
| mPlug-owl-7B | 37.60 | 27.58 | 50.40 | 34.22 | 40.78 | 28.97 | 50.00 | 33.33 | 41.77 | 29.46 | 44.11 | 30.71 |
| miniGPT-v2-7B | 58.25 | 57.59 | 65.30 | 64.33 | 50.74 | 48.44 | 51.04 | 35.97 | 53.53 | 52.65 | 55.77 | 51.80 |
| PromptHarm-GPT3.5 | 64.05 | 51.57 | 69.30 | 67.81 | 61.88 | 57.89 | 58.51 | 55.08 | 61.71 | 53.44 | 63.28 | 57.16 |
| Pro-Cap-GPT3.5 | 52.40 | 51.80 | 61.56 | 58.57 | 48.56 | 47.15 | 54.90 | 54.62 | 57.06 | 56.83 | 54.90 | 53.79 |
| CapAlign-GPT3.5 | 63.70 | 51.60 | 68.60 | 66.81 | 61.51 | 54.76 | 58.08 | 52.67 | 60.77 | 51.04 | 62.53 | 55.38 |
| Proposed | **68.40*** | **62.73*** | **73.10*** | **73.04*** | **62.18*** | 58.96 | **70.11*** | **69.97*** | **63.23*** | 61.25 | **67.40*** | **65.19*** |

- *Harmful* content can be defined as speech or material that mocks or ridicules a targeted person or organization or has the potential to cause emotional discomfort to any individual, politician, celebrity or the general public.

**Baselines:** We compare our method with recent SOTA LMMs and caption-based methods designed for meme analysis in zero shot settings. For the LMMs, we used the following: LLaVA [31], CogVLM [43], Qwen-VL [4], InstructBLIP [12], miniGPT4 [50], OpenFlamingo [3], MMGPT [17], Fuyu[4], mPLUG-Owl [49], MiniGPT-v2 [10] and LLaVA-1.6-13b [30]. For the caption-based methods, we use PromptHarm [19] Pro-Cap [7] and CapAlign [18]. To align with our zero-shot setting and for a fair comparison, for all caption-based baselines, we used GPT-3.5-turbo as the classification model.

## 5 Experimental Results and Analysis

### 5.1 Comparison with Baselines

Table 2 presents the overall performance of our proposed approach when compared to SOTA methods across five tasks related to social abuse in memes. We can see from the results that the performance of LMMs is relatively low in tasks that require reasoning, e.g., sarcasm detection. It is worth noting that many results showed an accuracy of 50% and an F1 score of 33.3% or close, indicating that the model incorrectly classified all memes as either "sarcastic" or "not sarcastic". This suggests a significant deficiency in the model's ability in this aspect. In some more general tasks, such as misogyny and Hatefulness, these models typically perform well because these kinds of tasks do not require too much reasoning.

The results reveal notable variations in model performance, with specific models like LLaVA-1.5 and Qwen-VL demonstrating better adaptation to the complexity of multimodal meme tasks than others. Interestingly, models with more parameters, such as CogVLM and MMGPT, do not consistently outperform others across all tasks. Furthermore, we observe that none of the SOTA LMMs perform well across all tasks designed for different tasks, making them less desirable for our specific task. These insights emphasize the intricacies of multimodal meme understanding and underscore the necessity

for improved methods to enhance performance in identifying social abuse within memes.

For current SOTA caption-based methods that are designed especially for meme analysis, the overall performance for PromptHarm is better than the LMMs, as expected. This is because the prompts generated were usually clear, although not detailed enough sometimes, especially when the meme images were complicated. We observe that Pro-Cap did not perform well in most tasks, except for Hatefulness, which the probing-based caption questions were initially designed for. In other tasks that do not match these questions, this method fails. We also observed that CapAlign, trained on the *Harmfulness* category, can perform well on harmful memes. However, it fails under zero-shot conditions on memes and is not generalizable (i.e., it does not perform well on memes other than harmfulness). We attribute this phenomenon to its uncontrolled automatic question-asking process, where, in some cases, the key information cannot be correctly extracted. Our method allows the model to constantly summarize the existing information during the dialogue process, raising effective questions. This way, the final number of question-answer pairs obtained is far less than CapAlign, but the quality is higher.

Our method outperformed all tested baselines across most tasks, including LMMs and methods specifically designed for memes. It achieved an overall F1-score of 65.19% (an increase of 7.59%) and an overall accuracy of 67.40% (an increase of 4.12%) compared with the second-best results. Our method is robust and stable, showing no significant performance gap when faced with different tasks.

### 5.2 Analysis

**Ablation analysis:** Through an ablation analysis (see Table 3), we determined that both components of our method contribute to its overall performance to varying degrees. Interestingly, adding an analysis step before making a final judgment marginally improved performance, consistent with findings from previous studies [28]. Unlike complex tasks such as mathematical reasoning, meme classification typically does not require many incremental steps. However, it relies on understanding image and text content and real-world information to make judgments. We observed a decline across all

---

[4]https://huggingface.co/adept/fuyu-8b

**Table 3: Ablation analysis of proposed framework without (w/o) analysis and w/o definition. * in ablation analysis indicates that proposed framework achieved a significant ($p < 0.05$) performance improvement over variants of the proposed framework under Mann–Whitney U test.**

| Model | Hatefulness | | Misogyny | | Offensiveness | | Sarcasm | | Harmfulness | | Overall | |
|---|---|---|---|---|---|---|---|---|---|---|---|---|
| | Acc. | F1 | Acc. | F1 | Acc. | F1 | Acc. | F1 | Acc. | F1 | Acc. | F1 |
| Proposed | **68.40*** | **62.73*** | **73.10*** | **73.04*** | **62.18*** | **58.96*** | **70.11*** | **69.97*** | **63.23*** | **61.25*** | **67.40*** | **65.19*** |
| Proposed w/o analysis | 67.60 | 62.17 | 71.90 | 71.69 | 61.51 | 58.20 | 69.04 | 69.49 | 61.55 | 57.73 | 66.92 | 64.26 |
| Proposed w/o definition | 67.85 | 62.10 | 70.70 | 70.66 | 60.57 | 57.98 | 69.51 | 69.51 | 60.56 | 58.64 | 65.84 | 63.98 |

tasks when removing the definition, suggesting its utility in better understanding the task, especially in zero-shot settings for our method and the LMM baselines. Therefore, we conclude that our method's effectiveness lies in integrating all modules, collectively contributing to improved performance.

**LLaVA as vision expert:** The performance of LMMs alone in previous study [28] reveals a significant disparity between their general image understanding and meme classification abilities in a zero-shot manner despite their proficiency in describing provided images. While models like LLaVA-1.6 and CogVLM exhibit strong image understanding and instruction-following capabilities, they struggle with poor zero-shot performance in meme classification. To address this limitation, we conducted extensive experiments with CogVLM as the vision expert in our proposed method, showcasing the versatility and effectiveness of our approach (Figure 3). The comparison between LMMs alone, LMMs with added definition, and our multi-agent structure demonstrate our method's generalizability and ease of implementation for real-world tasks. This plug-and-play nature underscores the broader applicability and superior performance of our approach beyond specific model limitations.

**Analyzing the effectiveness of adding definition to LMMs:** In our proposed framework, incorporating the definition resulted in an overall increase and demonstrated consistent improvements across all tasks. However, when examining the impact of the definition on other well-performing LMMs, such as CogVLM, LLaVA (both 1.5 and 1.6), and Qwen-VL, the results varied (Figure 4). This inconsistency in performance enhancement across different models can be attributed to the LMMs' limited semantic comprehension and inability to extract information from longer contexts. As such, leveraging

LLMs for this task is more reasonable, given their superior ability to reason and follow instructions. Moreover, adding the definition provides flexibility akin to the training process. However, it is more adaptable to a broader range of situations beyond the constraints of the current dataset, making it a valuable tool in a zero-shot setting.

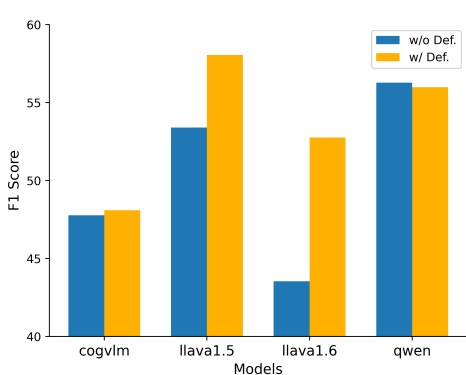

**Figure 4: Baselines' overall performance (F1 Scores) with (w/) and without (w/o) definition (def.) of tasks.**

**Quality check of randomly sampled QA pairs:** We further evaluate Multi-agent Chat using cases from our dialogue history (see Table 4). From a random selection of ten instances per task, we gathered 50 instances comprising 191 QA pairs. These were labeled as either *Good* or *Average and below*, with repeated instances also noted. Analysis showed that 63.9% of the QA pairs were rated *Good* for effectively posing desired questions, demonstrating the agents' proficiency in asking informative questions. Additionally, 7.3% of the QA pairs exhibited slight repetition, often due to unclear or indirect responses from the vision expert, prompting the agent to rephrase or re-approach the question.

**Effectiveness of Multi-agent Chat:** To objectively demonstrate the effectiveness of Multi-agent Chat, we present representative examples from the summary section alongside their initial descriptions across various task categories, which indirectly reflect the quality of the chats. As illustrated in Figure 5, Multi-agent Chat provides more

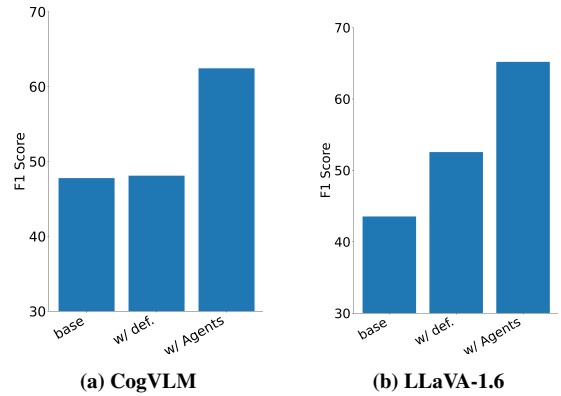

**(a) CogVLM**     **(b) LLaVA-1.6**

**Figure 3: Comparison of LMMs (CogVLM and LLaVA-1.6) as vision expert in our method. 'base' refers to the base LMM model, 'w/ def.' refers to adding definition to the prompt of LMM model, and 'w/ agents' refers adding multi-agent module.**

**Table 4: Quality check of randomly sampled QA pairs. Good quality and repetition ratio is evaluated. Note: informative QA pair is labeled as 'Good', and repetition means the QA is similar to previous round.**

| Category | Hate. | Miso. | Offen. | Sarc. | Harm. |
|---|---|---|---|---|---|
| Good (%) | 65.0 | 55.3 | 63.6 | 72.5 | 62.5 |
| Repetition (%) | 5.0 | 7.9 | 12.1 | 5.0 | 7.5 |

| Analysis Examples | | |
|---|---|---|
| **Meme** | **Description generated using LLaVA** | **Description generated using our method** |
| 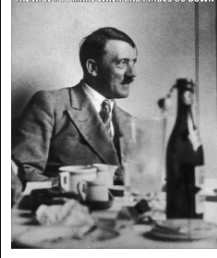 | The image is a black and white photograph of a man with a stern expression, wearing a military uniform with a tie. The text overlay on the image reads "when you get in trouble for eating all the cookies." This meme is typically used to convey a sense of humor or irony, suggesting that the person in the photo is being punished for a minor offense, **such as eating all the cookies**, which is a playful way to describe a situation where someone is **being held responsible** for a trivial matter. | Based on the information provided, the meme itself does not contain any explicit hateful content targeting ethnicity, gender, disability, or any other specific group. The text overlay on the image, "when you get in trouble for eating all the cookies," is a light-hearted and humorous scenario that does not appear to be intended to cause emotional discomfort or target any particular group. The man in the **military uniform**, although associated with a **German nationality** and a **stern expression**, is not depicted in a hateful context within the meme''s intended playful interpretation. |
| | The image is a black and white photograph of **a man** sitting at a table with various items on it, including cups and a bottle. The man is wearing a suit and tie, and he appears to be looking slightly to his right with a **neutral expression**. The text overlay on the image reads "THE FACE YOU MAKE WHEN GAS PRICES GO DOWN." This suggests that the image is being used to convey a humorous or satirical message about the **emotional reaction to a decrease in gas prices**, implying that the man''s expression is one of relief or satisfaction. | The meme in question uses the image of **Adolf Hitler**, a historical figure associated with atrocities and hate, to convey a message about the reaction to **gas prices going down**. The juxtaposition of Hitler's image with a **seemingly mundane** topic like gas prices can be seen as trivializing or **mocking serious historical events** and the impact of Hitler's actions. This could potentially cause emotional **discomfort** to individuals due to the use of a figure linked to hate and oppression in a lighthearted context. The meme''s content may be considered hateful as it involves using a symbol of hate in a humorous or satirical manner, which could be perceived as dehumanizing and insensitive. |

**Figure 5: Comparison between the description generated from LLaVA-1.6 and Our method. Incorrect or biased information is highlighted in Red, and good ones are in Green.**

comprehensive and unbiased information compared to the original descriptions. By thoroughly understanding the content of the memes, GPT effectively filters and integrates relevant information, leading to more accurate predictions.

Moreover, Figure 5 highlights that the descriptions generated using our method not only capture the visual details of the images but also integrate rich contextual information related to celebrities and cultural elements. For example, the system can recognize and reference figures like *Hitler*, identifying attributes such as *military uniforms* and *German nationality*. This demonstrates the model's ability to integrate knowledge from diverse cultural and historical backgrounds, enhancing the depth and relevance of the descriptions. This capability our our method significantly aids in assessing the potential for social abuse in memes, particularly when addressing sensitive or globally varied topics[5].

**Proposed Framework with Smaller Models:** We further evaluated the performance of the proposed framework by replacing the large models with smaller ones, specifically using Vicuna 7b as the agent and LLaVA1.6-7b as the vision expert, and compared these to the baseline LLaVA1.6-13b model. The results, as illustrated in Figure 6, demonstrate that our framework achieves an improvement of more than 10 percentage points in overall F1 score compared to the 13b LLaVA model. This notable improvement shows the effectiveness of our approach, even when smaller models are used.

**Case Study:** We selected two samples from our final result to analyze our proposed method further (see Figure 7). The left Meme (Sample 1) was correctly classified as Sarcastic, and the right Meme (Sample 2) was a wrong sample from Misogynistic classification.

---

[5]The flexibility of our framework allows for interchangeable vision and LLM components, enabling the integration of more culturally aware models to enhance the system's ability to detect social abuse across diverse cultural contexts.

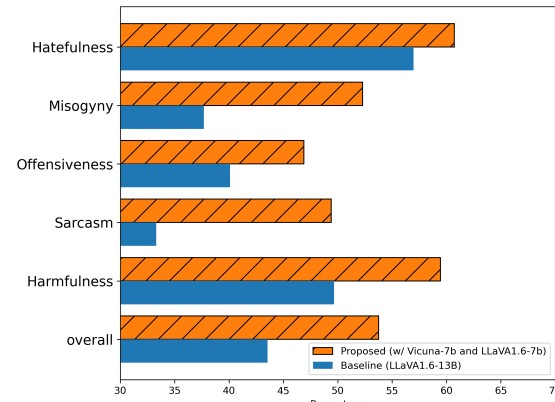

**Figure 6: Proposed Framework with Smaller Models: We replaced the large models with smaller ones i.e., used Vicuna 7b as the agent and LLaVA1.6-7b as the vision expert, and compared with the baseline LLaVA1.6-13b model**

Unimportant information has been removed from the texts for a better demonstration.

The correct sample on the left shows that LLaVA, as the vision expert, can extract meaningful information and fully understand the image even if the logo of McDonald's is not complete. Questions from two agents also show great relevance to the meme text in Sample 1, as they all focus on the health and quality of the food inside the image. Although the QA history shows that the overall quality of the food is good, the model still considered information that cannot be derived from the image alone or provided by the vision expert and made a correct judgment. This ability is extremely helpful for memes, as the information inside the image is normally limited and can also be biased.

| | | Sample 1 - Correct | | | | Sample 2 - Incorrect | | |
|---|---|---|---|---|---|---|---|---|
| Meme |  | feeding my abs nothing but the best quality beef | GT Label | **Sarcastic** |  | DON'T BRING A GIRL TO THE NBA FINALS SHE'LL GET BORED. | GT Label | **Not Misogynistic** |
| | | | Pred Label | **Sarcastic** | | | Pred Label | **Misogynistic** |
| Description | The image shows two **McDonald's** hamburgers and a portion of french fries in a paper container. The hamburgers are topped with ... The fries are … | | | | The image is a meme featuring two panels with text and photos. The top panel shows ..., appearing to be in a state of surprise or shock. The bottom panel shows ... The meme is likely making a humorous commentary on the idea that attending an NBA game with a girlfriend might not be as engaging... | | | |
| 1st round | Q: What is the appearance and quality of the McDonald's hamburgers in the image? | | | | Q: What are the **facial expressions** of the woman and man in the meme? | | | |
| | A: The McDonald's hamburgers in the image appear to be **freshly** prepared and have a **golden-brown** bun with sesame seeds on top. | | | | A: The woman in the meme has a surprised or shocked expression, while **the man** has a neutral or slightly **bored** expression. | | | |
| | Q: What is the overall presentation of the meal in terms of portion size and arrangement? | | | | Q: Is there any indication that suggests the woman's surprise is specifically related to attending the NBA Finals? | | | |
| | A: The meal consists of **two large** hamburgers and a **portion** of french fries. | | | | A: Yes, the image suggests that … | | | |
| 2nd round | Q: Are there any visible condiments or sauces on the hamburgers in the image? | | | | Q: Are there any other elements that reinforce the idea that the NBA Finals would be boring for a girl? | | | |
| | A: No, there are no visible condiments or sauces on the hamburgers in the image. | | | | A: Yes, the image shows a woman … , and **a man** looking at his phone, which might suggest that **he is not** fully **engaged** in the event. | | | |
| | Q: Is there any visible branding or logo on the packaging of the McDonald's hamburgers and fries in the image? | | | | Q: Is there any element that challenges or contradicts the stereotype being perpetuated about women not enjoying the NBA Finals? | | | |
| | A: Yes, there is visible branding on the packaging. The packaging has the McDonald's **logo**. | | | | A: The image shows … | | | |
| Analysis | Considering the meme text "feeding my abs nothing but the best quality beef" paired with the image of McDonald's hamburgers and fries, it can be interpreted as sarcastic. The phrase "best quality beef" suggests a level of health consciousness and a focus on consuming high-quality, nutritious food for physical fitness, which is ironic given that fast food like McDonald's is often associated with being unhealthy and not typically considered high-quality or nutritious. | | | | Based on the provided information, the meme perpetuates the stereotype that women would not enjoy attending the NBA Finals and suggests that bringing a girl to such an event would lead to boredom. This perpetuation of gender stereotypes and the assumption that women would not find sports events engaging can be considered as a form of stereotyping and potentially objectification. | | | |

**Figure 7: Case study of both correct (left) and incorrect (right) predictions of our method. Important information is in Bold.**

From the incorrect Sample 2 of Misogyny, we can also see some shortcomings of our method. The analysis still seemed to revolve around the woman's actions in the image and ignored the male character. There was also a certain degree of repetition of the questions, resulting in an excessive output of useless, redundant content.

## 6 Ethics and Broader Impacts

Our study ensures reproducibility through detailed experimental setups and parameter configurations provided in Section 4, facilitating transparency and replication of results with the availability of code[6]. Regarding user privacy, we prioritize protecting personal data by using information that does not involve sensitive user information and crediting dataset sources for copyright compliance. While we have taken measures to mitigate biases, unintentional biases may exist due to publicly available datasets, underscoring our commitment to impartial analysis and respect for all individuals and communities. However, there is potential for misuse of our method in spreading false information, necessitating supervision to prevent misuse. Our adherence to the intended usage of publicly available data and open-source platforms for research purposes underscores our commitment

---

[6]Anonymized following the double-blind guidelines

to ethical research practices. Nevertheless, the broader impact of AI models in auditing social media platforms raises ethical questions regarding the standard of harmfulness and AI's role in judgment, highlighting the need for further research to address these moral debates and emphasizing the importance of human oversight in AI operations for social abuse meme detection.

## 7 Conclusion

This paper introduces a robust multimodal approach to generating informative visual descriptions of memes. By employing insightful questioning within a zero-shot visual question-answering framework, we leverage the capabilities of GPT-3.5 as dual agents and LLaVA-1.6 to extract high-quality information related to potential social abuse in memes. Through iterative discussion and information acquisition, our method ensures comprehensive analysis. Our experimental results on the GOAT-Bench dataset, comprising over 6K memes across various social abuse tasks, demonstrates superior performance than SOTA methods. This underscores the efficacy of our approach in enhancing meme analysis and its potential to contribute to detecting social abuse on online platforms.

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

## A  Appendix

### A.1  Prompt Template of Agents

To fully utilize the role-play capability of LLMs, we designed a general prompt template for social abuse detection in memes. For agent involved in our proposed framework, system prompt (Table 6(a)) and user prompt (Table 6(b)) are concatenated as inputs for LLM in the *Discussion Process* and Table 5 is for the *Summary and Classification module* described above.

**Table 5: Summary and Classification**

**[System]**
*You are an AI assistant tasked with analyzing a meme to determine if it can be considered [adj].*
*You will be provided with the following information:*
*<information>*
*### 1. The meme's detail, including its description, the meme text.*
*### 2. QA history that helps understand the meme's content and context.*
*</information>*

**[User]**
*{Initial information, refer to Table 6}*
*This is the QA history:*
*### Question: {Quesiton}*
*### Answer: {Answer}*
*\* n rounds*

*This is the definition of [adj] content:*
*<definition>*
*[Definition of the task here]*
*</definition>*

*Please follow these steps:*
*### Step1: Based on fully examination of the provided information, please analyze how the meme could be considered [adj] or not.*
*### Step2: Provide a definitive answer of either "YES" or "NO" to indicate if the meme is [adj] or not in the format:*
*<answer> your answer </answer> based on your analysis.*

**Table 6: Prompt for Agents**

**((a)) System prompt for Agents**

[System]

*You will be assigned a specific role in a discussion to determine whether a particular meme is [adj] or not.*

*There is another referee assigned the same task, it's your responsibility to discuss with him and think critically before you make your statement.*

*You should respond according to the instructions and follow the format given in the example:*

*<instructions>*

*### 1. Your task is to maximize your information about a given meme's image content in order to finally determine whether the meme is [adj] or not.*

*### 2. Please don't talk about anything that is not related to the meme and make your statement short and concise.*

*### 3. Don't copy answers from the previous discussions.*

*### 4. You can pose one basic question about the image's content only, and it should be different from the questions already asked.*

*### 5. Your question should be simple and directly related to the content of the image that can help you analyze.*

*</instructions>*

*<example>*

*[your analysis should be here]*

*<question> [your question about the image's content should be here] </question>*

*</example>*

*You will be given the meme's information, the history of discussions, including questions/answers from yourself and another person in this task.*

**((b)) User prompt for Agents**

Init information:

*This is the meme's information:*

*<information>*

*The meme's description: {initial caption from vision expert tool}*

*The meme's text: {text}*

*</information>*

Discussion history:

*This is the discussion history:*

*<history> {chat history from previous rounds} </history>*

Role definition:

*You are now [role], one of the referees in this task. You have spent a lot of time on social media and have seen many memes of different types.*

*You should follow the given instructions to assess whether a particular meme is [adj] or not, but do not make any definitive judgment.*

*Please make your point short and clear based on the meme's information and discussion history and critically thinking, and then ask a question about the meme image in the format <question>your question</question>.*

*Now it is your time to talk, [role]:*

