# OpenReview forum: "Ask, Acquire, Understand: A Multimodal Agent-based Framework for Social Abuse Detection in Memes"
_ACM.org/TheWebConf/2025/Conference — WWW 2025 Poster_

### Official Review · Reviewer_jRrB · 2024-11-21

**Novelty:** 4
**Technical Quality:** 5

**Review:**

**Quality**

- The paper presents a multimodal, agent-based framework that leverages large language models (LLMs) and a large multimodal model (LMM) for detecting social abuse in memes. The quality of the work is above average in terms of integrating different large models to address a complex problem.
- The experiments on a benchmark dataset sourced from diverse meme datasets demonstrate the framework's effectiveness.

**Clarity**

-	The paper is well-structured, with a clear abstract, introduction, methodology, experimental results, and conclusion.
-	The figures and tables are clear and support the textual content effectively.
-	In the introduction and the method sections, the deep motivation to apply or design the method is not presented clearly.

**Originality**

-	The paper's originality lies in integrating LLMs with LMM for social abuse detection in memes.

**Significance**

-	The work is significant, given the increasing need for tools to detect and mitigate social abuse in digital media, particularly in the context of memes which are prevalent on social platforms.
-	The framework's zero-shot learning capabilities are significant for practical applications where labeled data may be scarce.

**Pros**

-	The task “social abuse detection in memes” is of high practical significance and application value. Many previous methods also focus on this task.
-	Authors propose to leverage the zero-shot inference abilities of large models to address the task, which may benefit the generalization ability of the method. In addition, the paper notices the hallucination problem of generation models and try to alleviate its impact by applying certain techniques.
-	The paper delivers a thorough evaluation of the proposed framework, encompassing performance comparisons and presenting detailed, illustrative examples. This comprehensive assessment serves to validate the effectiveness and utility of the framework in question.

**Cons**

-	The innovation in the framework’s design is limited. The conversational agents system of LLM proposed by previous work [A, B] can form complex frameworks to solve a series of problems [C]. The paper leverages collaborations of multiple large model agents to solve the task. However, the most effective work process comprises a two-round discussion between conversational agents. This particular design does not seem to target the specific challenges inherent to the task.

[A] Li, Guohao, et al. "Camel: Communicative agents for" mind" exploration of large language model society." NIPS2023.

[B] Wu, Qingyun, et al. "AutoGen: Enabling Next-Gen LLM Applications via Multi-Agent Conversation." ICLR 2024.

[C] Shao, Yijia, et al. "Assisting in Writing Wikipedia-like Articles From Scratch with Large Language Models." NAACL 2024.

-	In both the introduction and methodology sections, the profound connection between the applied or designed method and the specific task is not adequately articulated. This relationship is crucial for understanding how the chosen approach directly addresses the objectives and challenges of the task, and it should be clearly delineated to provide a solid foundation for the research.
For instance, while the second paragraph of the introduction succinctly outlines the challenges associated with the task, it fails to establish a clear correlation between these challenges and the innovative aspects of the proposed frameworks that could potentially address them. This linkage is essential for demonstrating how the novel contributions of the research directly respond to the identified difficulties.
-	Based on the experimental results, the framework outperforms the list of baselines, yet its effectiveness is questionable for several reasons. Firstly, the ablation analysis compares settings that focus on minor aspects of the workflow, neglecting the validation of the multi-round conversation module, which is central to the framework. Secondly, replacing large models with smaller ones results in a significant performance decline of approximately 10%, suggesting that the performance gains are more attributable to the increase in model parameters than to the framework's design. Thirdly, the multi-round conversations showcased in Figure 7 do not align with the objective of 'acquiring as much image information about the meme as possible for the final social abuse detection.' This discrepancy raises concerns about the framework's ability to achieve its intended goals.

**Conclusion**

For the above reasons, I suggest *rejecting* the submission.
I recommend that the authors significantly revise the manuscript to address the issues, potentially including improving the framework design, a more thorough experimental evaluation, a detailed analysis of the model's decisions, etc.

**Questions:**

-	The paper utilizes the gpt-3.5-turbo, what is the cost analysis with respect to detection?
-	What is your methodology for hand crafting prompts to ensure the framework’s success rate?

**Reviewer Confidence:**

4: The reviewer is certain that the evaluation is correct and very familiar with the relevant literature

**Scope:**

4: The work is relevant to the Web and to the track, and is of broad interest to the community

---

### Official Review · Reviewer_nqum · 2024-11-24

**Novelty:** 6
**Technical Quality:** 6

**Review:**

This works proposed a multimodal and multi-agents framework for social abuse memes detection. This works notably utilize multi-agents ability to adaptively detecting the abuse memes without the human interactions. Other than this, the proposed method adopts a vision expert as a reference for the agent to enquire more image information. This enquire may process for several round to refine and correct the bias between the discussion of agents. This approach is promising to replace human questioners. In addition, a summary agent is designed to summary the information across the discussion phrase and classify the memes. The proposed method is simple but technically sound from human behaviour. The definition is a good reference for LLM to define what the gold standard of memes. Many experimental results are significantly improving over other baseline models, which is a good sign to prove the effectiveness of the proposed method.

**Questions:**

As many agent are introduced, it will be good to compare the efficacy of the proposed method with other baseline methods.
The author mentioned that the number of discussion rounds are set to 2. It would be great to see if the performance increases with the discussion round.

**Reviewer Confidence:**

3: The reviewer is confident but not certain that the evaluation is correct

**Scope:**

4: The work is relevant to the Web and to the track, and is of broad interest to the community

---

### Official Review · Reviewer_LimB · 2024-11-26

**Novelty:** 4
**Technical Quality:** 4

**Review:**

This paper applies a multi-agent approach to detecting social abuse in memes and achieves state-of-the-art (SOTA) performance. The writing is well-done, but there are still some issues that need to be addressed:
## 1. Results in Table 2: Baseline Prompt vs. Benchmark Discrepancy
The baseline prompt results in Table 2 show a significant discrepancy compared to the benchmark results (e.g., GOAT). This discrepancy arises even though only the instruction "answer yes or no" was added. Could you provide further clarification on why such a difference occurs?

Additionally, the inclusion of multi-agent settings and definitions for each agent is quite common in related studies [relevant citation]. However, the paper specifies that only two rounds of discussion were conducted, without experimental evidence to justify this choice. What happens if the number of rounds is set to one? Or increased to three? Exploring these scenarios would strengthen the argument.

## 2. Complexity of the Proposed Method
The proposed method has high complexity, as it requires an initial captioning model, two discussion models, a vision expert model, and a final summary assistant model.

However, the benchmark results from [GOATBench] show that GPT-4V outperforms the results in this paper, with seemingly similar costs. Considering the multiple API calls to ChatGPT in the proposed method, along with the high input and output token count, the cost comparison does not appear favorable. Could you provide a more detailed comparison of computational overhead and efficiency?

## 3. Roles in GPT-3.5 Play
The paper introduces the roles of "Internet User" and "Internet Supervisor" for GPT-3.5 in the Play2 setting. However, the prompts provided in the paper do not clearly reflect the distinct characteristics of these roles. Furthermore, there is no experimental evidence to demonstrate the necessity or effectiveness of assigning these specific roles. Additional clarification or validation is needed to support this design choice.

## 4. Figure 6: Model Comparison
Figure 6 compares the results of using smaller language models and a 7B vision expert against a 13B vision model. The observation that the former outperforms the latter is unsurprising. However, the experiments do not explore alternative scenarios:

How would the results differ if GPT-3.5 agents were replaced with stronger models (e.g., GPT-4) or weaker ones (e.g., LLaMA)?
How would the results vary if the vision model were replaced with either a smaller or a larger one?
Such analyses would provide valuable insights into the robustness and scalability of the proposed method.

## 5. Clarity Issues in Figure 2 and Ablation Studies
Figure 2 is not clearly illustrated, making it challenging for readers to follow the architecture. Additionally, the description of the analysis step in the ablation studies (e.g., "w/o analysis") is ambiguous. For example, the statement:
"Adding an analysis step before making a final judgment marginally improved performance, consistent with findings from previous studies"
raises the question of whether the "analysis step" refers to the summary part in Section 3.3.

Although Figure 7 provides case studies of the analysis step, the methods and experiments sections fail to provide a clear understanding of how this step works. This creates confusion for readers. Could you provide a clearer explanation and align the figures and experimental descriptions accordingly?

**Questions:**

please refer to the mentioned questions in above

**Reviewer Confidence:**

3: The reviewer is confident but not certain that the evaluation is correct

**Scope:**

3: The work is somewhat relevant to the Web and to the track, and is of narrow interest to a sub-community

---

### Official Review · Reviewer_WDyK · 2024-12-01

**Novelty:** 4
**Technical Quality:** 4

**Review:**

This work proposes a multimodal framework based on a multi-agent system for Social Abuse Detection in Memes. The approach combines the LLMs and LMMs to propose a multimodal, agent-based framework that generates informative visual descriptions of memes by posing insightful questions, and realizes the identification of malicious content through a zero-sample visual question-and-answer setting. Specifically, the paper generates high-quality visual descriptions through iterative Q&A and discussion via two agents with different roles (“Internet User” and “Internet Supervisor”), in collaboration with a visual expert model, and ultimately utilizes LLM to classify malicious content. Experimental results show that the framework outperforms current state-of-the-art methods in terms of accuracy and F1 scores on multiple datasets, demonstrating its effectiveness and generalizability for the task of malicious content detection in social media. However, despite this, I believe that the innovation and workload of this work is relatively limited for the reasons elaborated below.

First, regarding the overall quality and workload, this paper introduces a multi-agent structure for Social Abuse Detection in Memes, combining LLM and LMM to achieve recognition, showcasing a certain level of novelty. It conducts comprehensive comparative experiments on the latest benchmark dataset, GOAT-Bench, along with ablation studies and case analyses, demonstrating superior performance in terms of accuracy and F1 score compared to existing methods. The content is thorough, with well-organized and aesthetically pleasing tables and figures, reflecting a meticulous approach. However, certain shortcomings remain, which, in my opinion, impact the overall quality assessment of this work. These are outlined below:
1. The correlation between the proposed solution and the limitations of the existing work cited in the article is vague and does not clearly articulate how the approach proposed in this paper addresses the existing shortcomings. The three main shortcomings identified in the article are over-reliance on image caption, over-reliance on prompt, and lack of generalizability of existing models. However, the  solution proposed in this work still relies on the capability of LLM itself and the design of prompt, compared to GOAT-Bench which only attenuates the problem of LLM's own illusion through multiple rounds of multi-agent conversations. As stated in the paper, all the prompts in this paper are manually designed, and they do not address the second shortcoming you raised. In addition to this, multi-agent multi-round conversations do not seem to address the third flaw of generalization. On the contrary, multiple rounds of conversation will further exacerbate LLM's hallucination problem and the information redundancy that this brings, and the generalizability enhancement doesn't seem to be focused on in this paper. This point is therefore the main cause of my low rating of the quality of this work.
2. The baseline setting of the experiment is not reasonable enough. In the comparison experiment in Table2, a lot of space is spent on the comparison of LMM models, and only three comparison works are listed which are closely related to this problem, which in fact cannot compare the advancement of the work proposed in this work compared with other works under the same problem. Instead, you should focus on the comparison of other related models under this problem. However, to the best of our knowledge, the most recent SOTA models for malicious content detection of terrier graphs are far from being enumerated in your comparison experiment. This significantly reduces the quality and reliability of the experimental part of the analysis.
3. There are spelling mistakes in the text, while some of the figures are a bit blurred. In the first paragraph of the Discussion Process section in 3.2, framework is spelled as farmwork. in addition to this, Figure 5 and Figure 7 are not clear enough, with obvious blurring visible when zoomed in, which has an impact on the overall aesthetics of the paper.

Secondly, in terms of the clarity of the paper, the article is well structured, including methodology, experimental setup and result analysis, which is logical, while the readability of the content is enhanced by graphs and charts (e.g., framework architecture diagrams, performance comparison tables, etc.). However, there are still some problems:

1. The case study section explains very limited content, one of the core of this paper is to address the low robustness of the current model under terrier map malicious detection, but the case study does not analyze this in any way, which leads me to not have a clear perception of the real improvement part of the model effect in this paper.
2. The experimental comparison section does not provide a clear comparison and improvement of the effectiveness of existing SOTA models. As far as I know, there are at least such as DisMultiHate, ExplainHM, and RGCL. therefore, the experimental part of the article does not make a very clear presentation and comparison in terms of the state-of-the-art of the methods proposed in this work.

In terms of originality, this work possesses a certain degree of originality by proposing the use of a multi-agent based framework to achieve the recognition of malicious content through a zero-sample visual quiz setting. This is indeed an approach that has not been used before in the related field, and also this paper uses detailed experiments to prove the effectiveness of the proposed approach. However, I think the originality of this paper is very limited. In essence, the multi-agents proposed in this paper are using existing techniques (GPT-3.5), only the answers of LLM are refined to some extent by manually designed polytheistic Q&A, and this solution is not sufficiently specific for this problem. In other words, for any domain, the use of multi-agent Q&A leads to a performance improvement of the model compared to direct Q&A. I don't think this improvement is very innovative.

For the significance as well, multi-round Q&A is an improvement over single-round Q&A for any problem. The improvement of this paper compared to GOAT-Bench is nothing but the use of multi-agent framework, so I think the significance of this paper is rather limited.

In summary, although the paper demonstrates highlights in the framework design and experimental analysis, its limitations are in the innovation of the methodology, the completeness of the comparative experiments, and the relevance to the current shortcomings of the problem.

**Questions:**

1. How does your proposed multi-agent solution address the three flaws identified in your paper? What is the relevance of this?
2. Why does your experimental baseline setup spend a large amount of time on the selection of LMMs instead of the relevant SOTA models that are currently available? Why were only three relevant baseline models selected?
3. Does the lack of clarity in the presentation of parts of the case study demonstrate that the methodology still does not address the lack of generalizability of current models?
4. Why was the number of discussion rounds set to 2 in the experimental setup? What was the rationale for this choice?

**Reviewer Confidence:**

4: The reviewer is certain that the evaluation is correct and very familiar with the relevant literature

**Scope:**

3: The work is somewhat relevant to the Web and to the track, and is of narrow interest to a sub-community